# optoPAD, a closed-loop optogenetics system to study the circuit basis of feeding behaviors

José-Maria Moreira[†], Pavel M Itskov[†‡], Dennis Goldschmidt[†], Celia Baltazar, Kathrin Steck, Ibrahim Tastekin, Samuel J Walker, Carlos Ribeiro*

Champalimaud Centre for the Unknown, Lisbon, Portugal

**Abstract** The regulation of feeding plays a key role in determining the fitness of animals through its impact on nutrition. Elucidating the circuit basis of feeding and related behaviors is an important goal in neuroscience. We recently used a system for closed-loop optogenetic manipulation of neurons contingent on the feeding behavior of *Drosophila* to dissect the impact of a specific subset of taste neurons on yeast feeding. Here, we describe the development and validation of this system, which we term the optoPAD. We use the optoPAD to induce appetitive and aversive effects on feeding by activating or inhibiting gustatory neurons in closed-loop – effectively creating virtual taste realities. The use of optogenetics allowed us to vary the dynamics and probability of stimulation in single flies and assess the impact on feeding behavior quantitatively and with high throughput. These data demonstrate that the optoPAD is a powerful tool to dissect the circuit basis of feeding behavior, allowing the efficient implementation of sophisticated behavioral paradigms to study the mechanistic basis of animals' adaptation to dynamic environments.
DOI: https://doi.org/10.7554/eLife.43924.001

*For correspondence:
carlos.ribeiro@neuro.
fchampalimaud.org

[†]These authors contributed equally to this work

Present address:
[‡]Pharmacology, Sechenov First Moscow State Medical University, Moscow, Russia

## Introduction

The ability to experimentally manipulate the activity of neurons with cellular resolution has revolutionized our understanding of how circuits generate behavior (*Luo et al., 2018*). This ability has gone hand in hand with an improvement in technologies allowing for the automated and quantitative analysis of behavior (*Anderson and Perona, 2014*; *Branson et al., 2009*; *Brown and de Bivort, 2018*; *Calhoun and Murthy, 2017*; *Egnor and Branson, 2016*; *Mathis et al., 2018*; *Wiltschko et al., 2015*). While most methods for analyzing behavior quantitatively rely on video recordings, alternative methods play important roles in neuroscience research (*Davis, 1973*; *McLean and Kinsey, 1964*). Such approaches have become especially important to study feeding in *Drosophila melanogaster* (*Murphy et al., 2017*; *Ro et al., 2014*; *Yapici et al., 2016*). We have recently established a new method for quantifying feeding behavior in the fly which relies on capacitance measurements (*Itskov et al., 2014*). The flyPAD allows automated, high-throughput, quantitative analysis of feeding behavior with high temporal resolution. This high temporal resolution enables the dissection of feeding behavior at the level of the motor pattern and the microstructure of feeding. Using this framework, we have shown that different circuit and molecular mechanisms impinge on food intake by modulating two key variables of the feeding microstructure: the probability of initiating a feeding burst and the length of feeding bursts (*Corrales-Carvajal et al., 2016*; *Itskov et al., 2014*; *Steck et al., 2018*; *Walker et al., 2015*).

At the neural circuit level, gustation plays a pivotal role in regulating food intake. Classically, gustation has been shown to allow animals to both detect suitable food sources as well as to reject harmful foods (*Dethier, 1976*; *Jaeger et al., 2018*; *Scott, 2018*; *Yarmolinsky et al., 2009*). A key feature of gustation when compared to olfaction or audition is that it requires the physical

interaction of taste organs with the sampled substrate. As such, gustatory information is outstandingly contextual and depends critically on the behavior of the animal as it actively explores substrates. Ideally, therefore, manipulations of circuit function during feeding should be tightly coupled to the ongoing behavior. Techniques allowing for the real-time analysis of behavior have enabled neuroscientists to trigger neuronal manipulation specifically when the animal is performing specific behaviors. We have previously used such an approach to show that in *D. melanogaster*, taste peg sensory neurons specifically control the length of yeast feeding bursts (*Steck et al., 2018*).

Here, we describe the design and implementation of such a high-throughput system allowing optogenetic manipulation of neurons in *Drosophila* contingent on the feeding behavior of the fly: the optoPAD. We show that the optoPAD system allows for the specific, bidirectional manipulation of sweet and bitter neurons thereby triggering or suppressing appetitive or aversive feeding behaviors. We furthermore demonstrate the ability of our system to implement dynamic as well as probabilistic stimulation protocols. These protocols significantly expand the scope of the optoPAD, allowing for complex experimental designs. These additions significantly extend the toolset available to study complex behaviors in high throughput in *Drosophila*.

## Results

### The optoPAD system

We set out to develop a system allowing the optogenetic manipulation of circuit activity in *Drosophila* conditional on specific aspects of its ongoing feeding behavior (*Figure 1A*). To develop such a device, it is essential to be able to measure specific parameters of feeding behavior in real time. We previously developed the flyPAD system, which reliably measures feeding behavior in high throughput using capacitive proximity sensors from two food sources (*Itskov et al., 2014*). To allow for optogenetic manipulation of neurons, we designed an LED board housing a high-power multicolor LED (four colors), as well as metal-oxide-semiconductor field-effect transistor (MOSFET) gates and current limiting resistors, that fits on top of the flyPAD arenas (*Figure 1B*).

In the original flyPAD system, the relevant aspects of feeding behavior are extracted by offline processing of the capacitance signal after the behavioral experiment. We took advantage of the low complexity of the capacitance signal paired with the real-time data processing capacities of the Bonsai data stream processing language (*Lopes et al., 2015*) to analyze feeding behavior of flies in real time. Bonsai is a powerful visual programming framework especially designed for the acquisition and online processing of complex data streams such as those generated during behavioral experiments. We focused on using Bonsai to extract the periods in which the fly was actively interacting with food ('activity bouts'), as we previously showed that the total time of these interactions correlated well with total food intake (*Itskov et al., 2014*). After a series of simple signal processing steps (*Figure 1C*), we could obtain the onset time of activity bouts in real time. In order to precisely control the LED illumination, we designed a control breakout board (a standard 32-arena flyPAD system requires three of these boards), each of which uses one microcontroller (Arduino Mega) which operates as an IO device controlling the gates of the MOSFETs. This included a power distribution circuit to distribute power to the 128 channels of the LEDs (32 LEDs x 4 colors) (blueprints available at https://github.com/ribeiro-lab/optoPAD-hardware). The Arduino Mega runs a standard Firmata software, allowing it to function as a digital general-purpose input/output (GPIO) board from within the Bonsai environment. This allowed us to write all the controlling software, including the finite state machine controlling the experiments for all of the 64 channels of the flyPAD, in Bonsai.

The resulting optoPAD system has the following dataflow (*Figure 1D*): The flyPAD system uses a capacitance-to-digital converter to measure the interaction of the fly with the food. The capacitance information is sent to a computer via the flyPAD mainboard, and the real-time Bonsai algorithm detects when the fly starts to interact with one of the food electrodes. The software sends a signal to one of the digital pins of the microcontroller, which in turn controls the opening of the MOSFET on the LED board, leading to the illumination of a predefined LED color channel and thereby corresponding activation/inactivation of genetically identified neurons (*Video 1*).

Importantly, five parameters of LED activation can be easily controlled using this software: which food source triggers LED activation; which LED color is activated; the delay between the detection of the initiation of an interaction with the food and light onset; the duration the LED remains

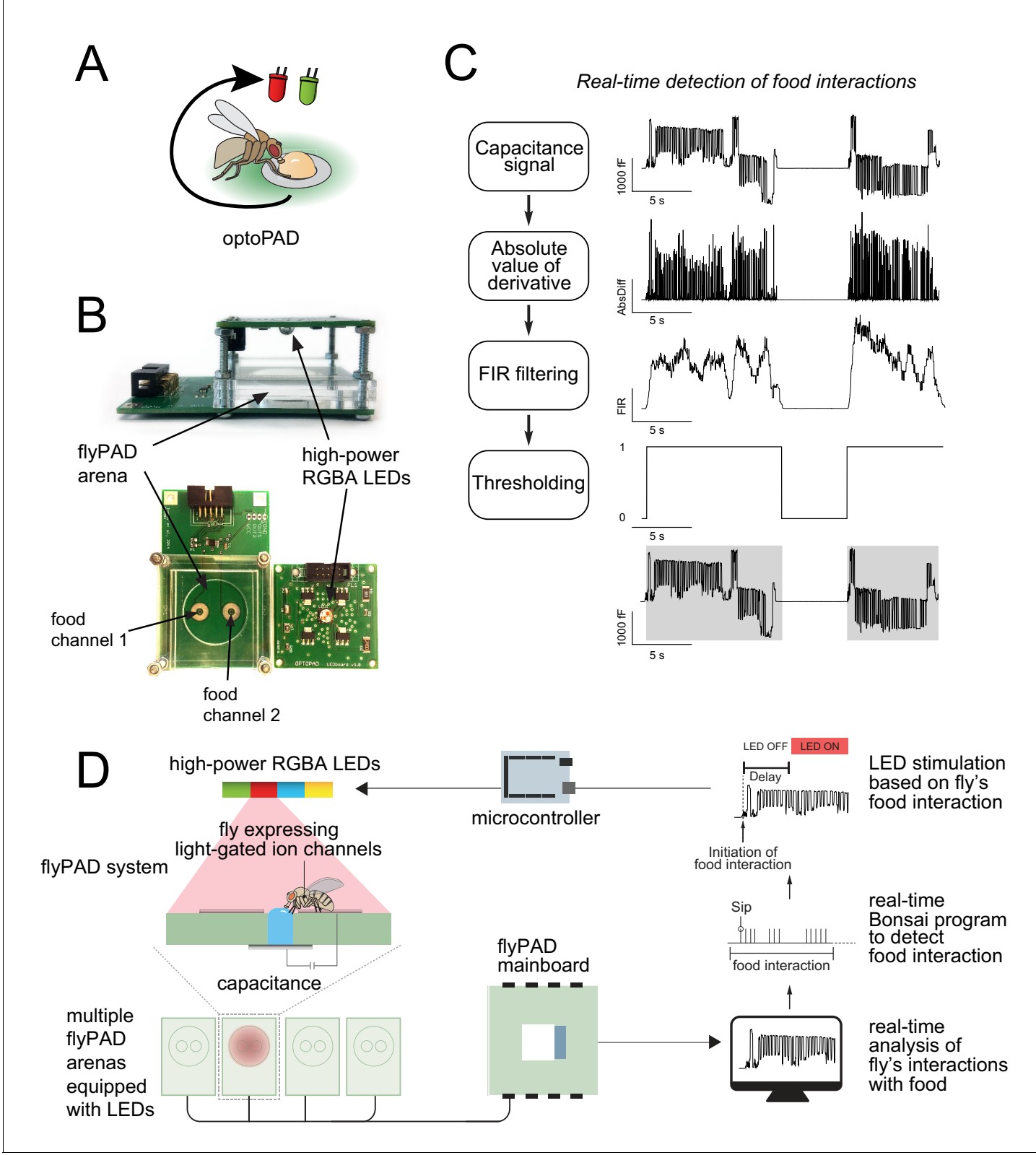

**Figure 1.** The optoPAD system. (**A**) Concept for the use of closed-loop capacitance measurement of feeding with optogenetic manipulation of neurons in behaving flies. The interaction of the fly with the food source triggers the activation of the LED. (**B**) Overview of the components of the optoPAD, the flyPAD arena and the high-power RGBA LEDs. (**C**) Algorithm for real-time detection of food interactions. Extracted food interaction bouts (activity bouts) are shaded in gray. (**D**) Schematic overview of the optoPAD experimental dataflow.

*Figure 1 continued on next page*

*Figure 1 continued*

DOI: https://doi.org/10.7554/eLife.43924.002

The following source data and figure supplements are available for figure 1:

**Figure supplement 1.** Quantification of the probability of the occurrence of a feeding burst onset depending on the time after activity bout onset.

DOI: https://doi.org/10.7554/eLife.43924.003

**Figure supplement 1—source data 1.** Source data file for *Figure 1—figure supplement 1*.

DOI: https://doi.org/10.7554/eLife.43924.004

on during the light stimulation; and the probability with which an onset of food interaction leads to LED illumination. Furthermore, the user can set how many times the light is triggered, can use a script which allows the delivery of the light after an activity bout is terminated, and can choose to deliver the light in an open-loop mode using predefined set intervals.

In order to ensure that the LED can be rapidly activated following the initiation of food interactions, we measured the latency of this system. The latency is defined as the time between food contact and LED illumination (with the 'delay' parameter set to 0). Our measurements revealed that it takes between 50 and 120 ms for the LED to be activated upon contact with the food. The latency is likely to originate in delays inherent to the serial communication bus, while the range of the latency is likely due to the buffer length of the flyPAD's system communication. Importantly, this latency is very short relative to the time it takes the fly to initiate a feeding burst after touching the food. This delay is quite reproducibly in the range of 400 ms (*Figure 1—figure supplement 1*). It is therefore very unlikely that the fly starts feeding before the LED is activated. By setting the delay parameter to 250 ms one can even time the light onset to coincide even better with feeding onset. Thus, our system is fast enough to 'close the loop' within the duration of a single sip.

To test how well the online activity bout detector works, we compared the performance of the new online algorithm with the previously validated offline algorithm (*Itskov et al., 2014*). We wrote a MATLAB script that replicates the online Bonsai workflow and classified each sample of a raw capacitance trace as belonging to an activity bout or not and compared this classification to the already validated flyPAD offline detector (*Itskov et al., 2014*). We then performed an ROC analysis to confirm the accuracy of the online detector. The detection of activity bouts by the online method correctly identified 91.5% of the capacitance trace samples as belonging to an activity bout (8.5% false negatives) and misclassified 1.6% of the samples as activity bouts (false positives). These data show that the online activity bout detector operates within the range of accuracy observed for state-of-the-art offline methods (*Itskov et al., 2014*).

## Optogenetic gustatory virtual realities

To validate the ability of the optoPAD system to manipulate neuronal activity in closed-loop, we decided to use it to create 'virtual gustatory realities' and test their impact on feeding behavior. To be able to better control the stimulation parameters, we used a switching DC power supply (40 A) to regulate the intensity of the LEDs. The luminous flux of the LEDs increased linearly with the forward voltage on the LEDs above 2 V for the red and amber LEDs and 2.5 V for the green and blue LEDs (*Figure 2A*), allowing us to vary the intensity of stimulation over a significant range. It is important to note that the maximum level of irradiance which can be achieved strongly depends on the wavelength of the LED.

To test if we can induce appetitive consummatory behaviors using the optoPAD, we used the *Gr5a-GAL4* line, which drives expression in sugar-sensing neurons of the labellum (*Marella et al., 2006*; *Thorne et al., 2004*). These neurons have been shown to be sufficient to initiate feeding (*Zhang et al., 2007*). Starved male flies expressing the red-shifted channelrhodopsin CsChrimson (*Klapoetke et al., 2014*) in

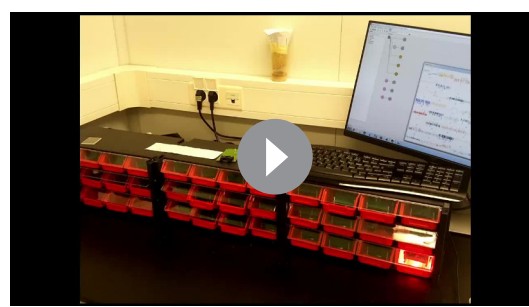

**Video 1.** Video showing the optopad system in operation.

DOI: https://doi.org/10.7554/eLife.43924.005

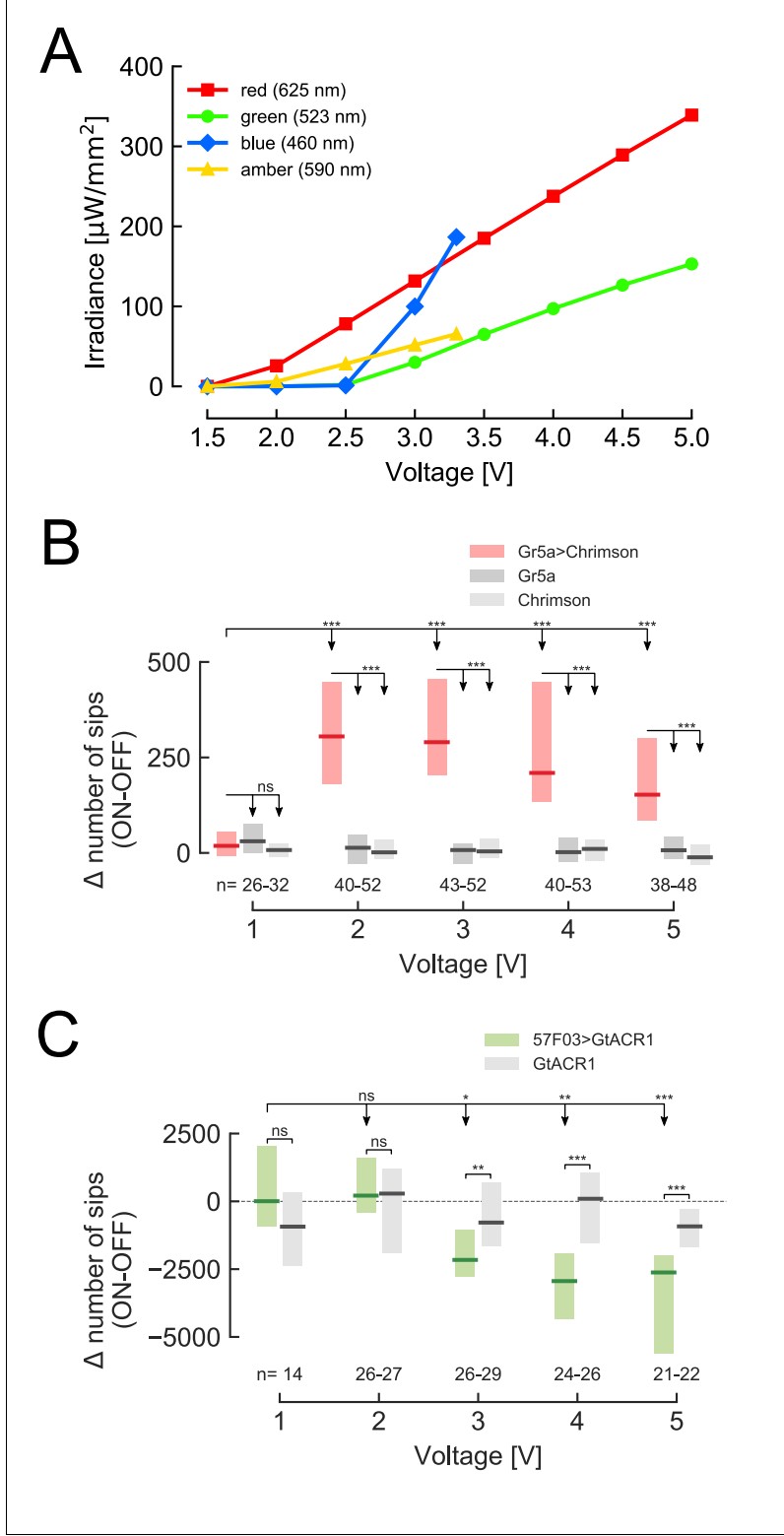

**Figure 2.** Increasing light intensity affects the feeding behavior of flies expressing different optogenetic effectors. (**A**) Irradiance of all four LEDs increases linearly with increasing voltage (for red and amber above 2 V, for green and blue above 2.5 V). The average value of the three measurements is shown and error bars indicate standard error of mean. (**B**) Difference in total number of sips on the stimulated (ON) and unstimulated (OFF) food patches of the same arena for 24 hr starved male flies expressing CsChrimson under the control of *Gr5a-GAL4,* and

*Figure 2 continued on next page*

*Figure 2 continued*

corresponding genetic controls. Both food sources contained 5 mM sucrose solution in 1% agarose. (**C**) Difference in total number of sips on the stimulated (ON) and unstimulated (OFF) food patches of the same arena for 3 days yeast-deprived, mated female flies expressing GtACR1 under the control of *57 F03-GAL4*, which labels taste peg GRNs, and corresponding genetic control. For genotypes, see Materials and methods and key resources table. Both food sources contained 10% yeast solution in 1% agarose. The numbers below the graphs indicate the number of flies tested in each condition. ***p<0.001, **p<0.01, *p<0.05, ns non-significance. Boxes represent median with upper/lower quartiles; groups compared by Wilcoxon rank-sum test, followed by Bonferroni multiple comparison test when more than two groups were compared.

DOI: https://doi.org/10.7554/eLife.43924.006

The following source data and figure supplements are available for figure 2:

**Source data 1.** Source data file for *Figure 2A*.
DOI: https://doi.org/10.7554/eLife.43924.011
**Source data 2.** Source data file for *Figure 2B*.
DOI: https://doi.org/10.7554/eLife.43924.012
**Source data 3.** Source data file for *Figure 2C*.
DOI: https://doi.org/10.7554/eLife.43924.013
**Figure supplement 1.** Mean number of sips per feeding bursts during open-loop stimulation (1 s ON, 2 s OFF) measured on both food patches of the same arena for 3 days yeast-deprived, mated female flies expressing *GtACR1* under the control of *57* F03-GAL4 or *67E03-GAL4*, which both label taste peg GRNs, and corresponding genetic control (*Steck et al., 2018*).
DOI: https://doi.org/10.7554/eLife.43924.007
**Figure supplement 1—source data 1.** Source data file for *Figure 2—figure supplement 1*.
DOI: https://doi.org/10.7554/eLife.43924.008
**Figure supplement 2.** Difference in total number of sips on the stimulated (ON) and unstimulated (OFF) food patches of the same arena for 3 days yeast-deprived, mated female flies expressing *CsChrimson* under the control of *SS02299-GAL4*, and corresponding genetic controls.
DOI: https://doi.org/10.7554/eLife.43924.009
**Figure supplement 2—source data 2.** Source data file for *Figure 2—figure supplement 2*.
DOI: https://doi.org/10.7554/eLife.43924.010

*Gr5a* neurons were given the choice to feed from two 5 mM sucrose sources in an optoPAD arena. One food source in each arena was programmed to trigger the activation of the red LED upon interaction (ON channel) while interactions with the other food source in the same arena never led to light activation (OFF channel). Even very low stimulation intensities (2 V) led to a clear and strong increase of feeding from the food source paired with optogenetic stimulation compared to the unstimulated source (*Figure 2B*). Interestingly, increasing the stimulation intensity did not lead to an increase in appetitiveness, indicating that a maximal behavioral impact can be achieved at low irradiances, thereby minimizing possible side effects caused by light.

Mated female flies deprived of protein for 3 days develop a robust appetite for yeast – their main protein source (*Carvalho-Santos and Ribeiro, 2018*; *Leitão-Gonçalves et al., 2017*; *Ribeiro and Dickson, 2010*; *Steck et al., 2018*). This appetite is driven by an increase in the length of feeding bursts, which is controlled by the activity of specific subsets of yeast gustatory neurons (taste pegs gustatory neurons) located on the labellum (*Corrales-Carvajal et al., 2016*; *Steck et al., 2018*). We previously demonstrated that silencing taste peg neurons with the anion channelrhodopsin GtACR1 (*Mohammad et al., 2017*) using the optoPAD system is sufficient to terminate feeding on yeast (*Steck et al., 2018*). To better characterize this silencing effect, flies were given the choice between two sources of yeast, one of which was paired with green light illumination. As we increased the intensity of light by increasing the voltage of the LED to 3 V, flies expressing GtACR1 in taste peg neurons fed significantly less from the channel paired with light compared to control flies (*Figure 2C*). In contrast to the *Gr5a* activation experiments (*Figure 2B*), the decrease in yeast feeding was accentuated with the increase in the applied voltage. This can be easily explained by the fact that the irradiance for the red and green lights reach the same intensity at different voltages (*Figure 2A*). Furthermore, it is important to note that in both experiments, we observed no behavioral effect of light in control genotypes. This indicates that the illumination itself has minimal effects on fly feeding behavior. Importantly, the optoPAD can also be used to stimulate neurons using

open-loop experimental designs. Indeed, silencing taste peg neurons by delivering the light in a fixed pattern not contingent on the behavior of the animal led to a specific decrease in the length of feeding bursts (*Figure 2—figure supplement 1*). Such experiments can be very useful when no hypothesis exists as to how the activity of a neuronal population of interest relates to the microstructure of the behavior. By correlating the relationship between light onset and specific behavioral phenotypes post-hoc, it should be possible to generate specific hypotheses of how the activity of a neuron relates to specific features of feeding behavior, guiding the design of specific follow-up closed-loop experiments.

To show that the optoPAD can also be used to manipulate sparse sets of neurons located deeply within the brain, we chose to activate giant fiber neurons in closed-loop upon initiation of feeding. The rationale of this proof-of-concept experiment was to use a sparse line labeling few (2) neurons deep in the brain of the fly, which has a readily observable phenotype (jumping), and which should have a clear effect on feeding behavior (termination of feeding). Indeed, upon the initiation of feeding and concomitant light activation, flies expressing CsChrimson in escape neurons (*Namiki et al., 2018*) jumped, leading to the termination of feeding (*Videos 2* and *3*). This led to a drastic decrease in feeding from the food source triggering light when compared with the control food source (*Figure 2—figure supplement 2*). These results show that the optoPAD can be used to manipulate very sparse, centrally located neurons, and monitor the impact of the manipulation on feeding behavior.

The mechanistic dissection of specific neurons' contribution to a behavior often requires the observation of opposite behavioral effects upon increases and decreases in their activity. Gustatory neurons are an ideal test case for this as they elicit both appetitive (e.g. sweet neurons) as well as aversive behavioral responses (e.g. bitter neurons). We tested the ability of the optoPAD system to both induce and suppress appetitive feeding responses using the *Gr64f-GAL4* line, which labels appetitive sugar-sensing neurons previously shown to be important to sustain carbohydrate feeding (*Jiao et al., 2008*). As observed for *Gr5a* neurons, closed-loop activation of *Gr64f* neurons using CsChrimson led to increased feeding (*Figure 3A*, left panel). This effect was absent in control genotypes (*Figure 3A*, right panel). Conversely, hyperpolarization of *Gr64f* neurons using GtACR1 led to a loss of appetitive behavior and hence a decrease in feeding from the sugar source paired with green light activation (*Figure 3B*).

To characterize the effect of closed-loop optogenetic manipulation of aversive neurons on feeding, we used *Gr66a-GAL4*, which labels bitter-sensing neurons (*Marella et al., 2006*; *Thorne et al., 2004*). In contrast to the depolarization of sweet gustatory neurons, flies expressing CsChrimson in bitter gustatory neurons immediately terminated feeding from an appetitive food source upon light activation (*Figure 3C*). This strong effect clearly mimics the potent aversive effect bitter substances have on feeding behavior. To test if we could suppress the aversive effect of bitter food using the optoPAD setup, we expressed GtACR1 in *Gr66a* neurons and observed the effect of green light activation on feeding from a quinine-laced sucrose solution. Indeed, flies exhibited higher feeding from the bitter food source paired with light stimulation than from the unpaired food source (*Figure 3D*). These experiments demonstrate the ability of the optoPAD to induce and suppress both appetitive and aversive effects on feeding behavior using

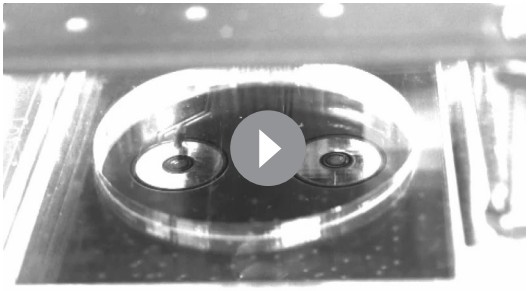

**Video 2.** Optogenetic activation of the giant fiber neurons marked by the C17-Gal4 line triggers escape responses upon feeding initation.
DOI: https://doi.org/10.7554/eLife.43924.014

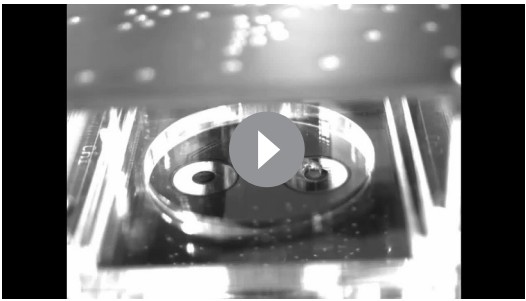

**Video 3.** Optogenetic stimulation of control fly triggers no escape responses upon feeding initation.
DOI: https://doi.org/10.7554/eLife.43924.015

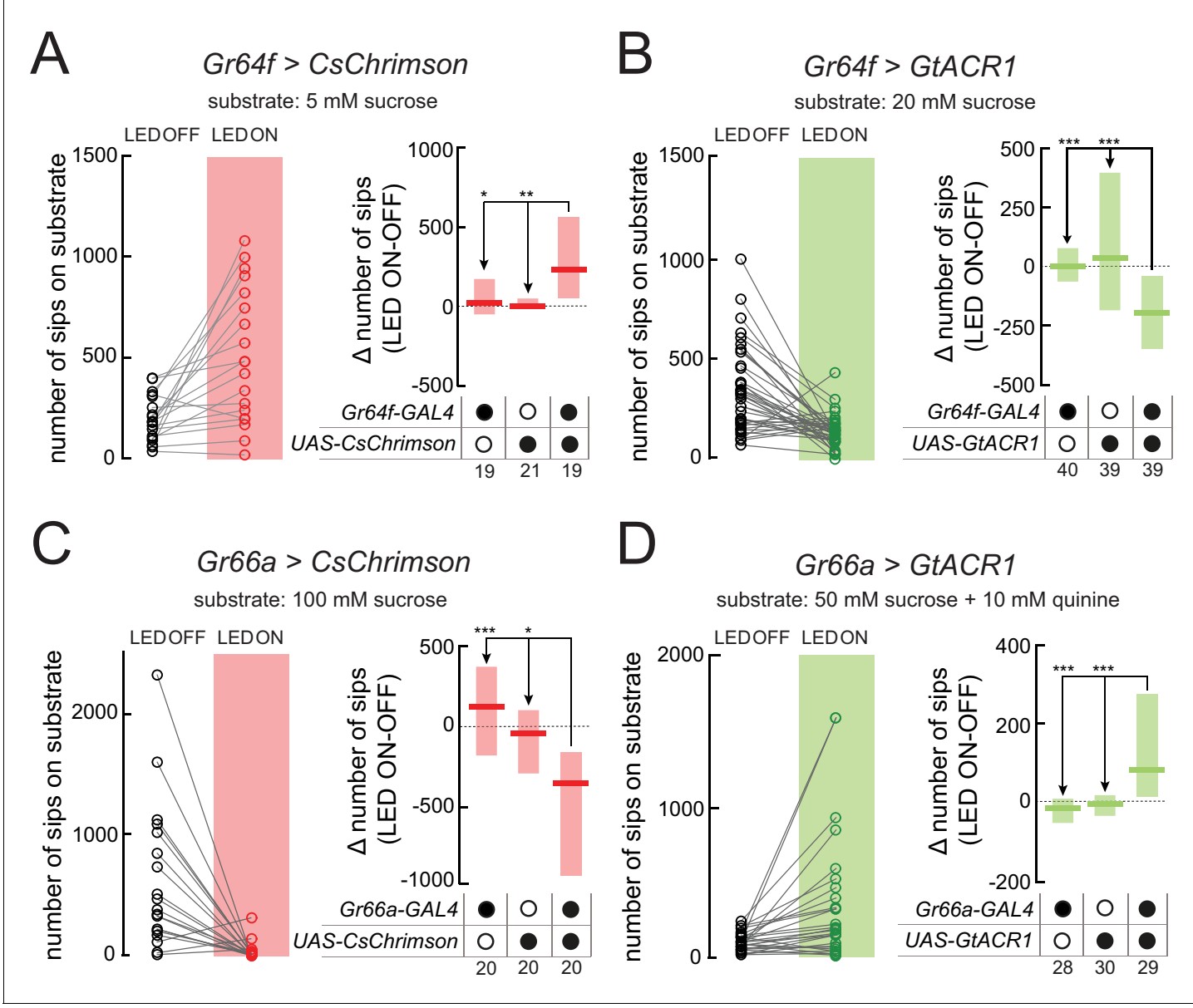

**Figure 3.** Creating virtual taste realities for *Drosophila* using the optoPAD. (A–D) Total number of sips from the unstimulated (LED OFF) and the light-stimulated (LED ON) food patches of the same arena by flies expressing CsChrimson (**A and C**) or GtACR1 (**B and D**), under the control of *Gr64f-GAL4* (**A and B**) or *Gr66a-GAL4* (**C and D**) (left side of the panels). Difference in total number of sips on the stimulated (ON) and unstimulated (OFF) food patches for flies expressing CsChrimson or GtACR1 (**A and C**), under the control of *Gr64f-GAL4* (**A and B**) or *Gr66a-GAL4* (**C and D**), and corresponding genetic controls (right side of the panels). All flies were 24 hr starved male flies (for genotypes, see Materials and methods and key resources table). The food substrate is indicated in each panel. The numbers below the graphs indicate the number of flies tested in each condition. ***p<0.001, **p<0.01, *p<0.05. Boxes represent median with upper/lower quartiles; groups compared by Kruskal-Wallis test, followed by Dunn's multiple comparison test.

DOI: https://doi.org/10.7554/eLife.43924.016

The following source data is available for figure 3:

**Source data 1.** Source data file for *Figure 3A*.
DOI: https://doi.org/10.7554/eLife.43924.017
**Source data 2.** Source data file for *Figure 3B*.
DOI: https://doi.org/10.7554/eLife.43924.018
**Source data 3.** Source data file for *Figure 3C*.
DOI: https://doi.org/10.7554/eLife.43924.019
**Source data 4.** Source data file for *Figure 3D*.
DOI: https://doi.org/10.7554/eLife.43924.020

either optogenetic activation or inhibition of different neuronal populations.

## Creating dynamic gustatory virtual realities

One of the unique features of virtual realities is the ability to generate stimuli that can change dynamically in a way that might be impossible with natural stimuli. The high temporal precision of optogenetics makes it ideal to both tightly link changes in activity of specific neurons to the behavior of the animal, as well as to change this contingency in a flexible while precise manner. We therefore implemented the ability to arbitrarily predefine the conditions upon which the behavior of the animal triggers light activation. As a proof of concept, we first set out to determine how flies would respond to dynamically changing the identity of the food source triggering gustatory stimulation. When flies expressing CsChrimson in bitter neurons (using *Gr66a-GAL4*) are given the choice between two identical appetitive food sources, they avoid feeding from the one paired with light activation (*Figure 3C*). Bonsai allows us to change the variables controlling this stimulation in a dynamic fashion. We programmed the system to switch the identity of the food source paired with light activation every 5 min (*Figure 4A*). From the beginning of the assay, the behavior of the flies appears to follow the stimulation pattern, with flies feeding less from the food source paired with light activation (*Figure 4B*). Only after 10 min of exploration, however, did these preferences reach statistical significance. As clearly visible in the raster plots of the feeding behavior, some flies always interact with both channels, but in the two first blocks of the experiment (0–5 min and 5–10 min) overall few flies interact with the food (*Figure 4—figure supplement 1*). This likely reduces the statistical power to detect a possible preference and might be due to an 'acclimatization' period from the moment the animals are introduced into the chambers (*Corrales-Carvajal et al., 2016*).

Next, we tested if we can alter the length of the interaction of the fly with food by increasing the delay between the initiation of food interactions and LED activation (*Figure 4C*). By initiating the optogenetic activation of *Gr66a* neurons 1.5, 3 or 6 s after the fly starts interacting with the food, we could make the flies terminate their interaction with food after precisely 2.66, 4.25 or 7.34 s, respectively (*Figure 4D*). This was a dramatic shortening of their activity bout, as control flies displayed reproducibly long bouts with a median of around 15 s. These experiments show that the optoPAD system can be used to dynamically change the contingency between the behavior of the animal and the optogenetic stimulation.

## Creating probabilistic gustatory virtual realities

While the optogenetic experiments described up to this point have been deterministic in nature, behavioral experiments in which the behavior of the animal is linked to a probabilistic delivery of a reward or punishment have been very powerful in probing the neuronal substrates of complex learned behaviors (*Fiorillo et al., 2003*). Such designs can be either used to allow the animal to learn specific statistical properties of the environment (*Lottem et al., 2018*), or unstimulated trials (catch trials) can be used as controls within a task (*Lak et al., 2014*). We tested the ability of the optoPAD system to implement such probabilistic experimental designs. We used Bonsai to set the probability of red light activation upon food interactions to 90% for both food sources (*Figure 4E*). Therefore, 10% of food interactions (trials) did not result in LED activation. Importantly, these 'catch trials' were randomly selected and therefore could not be predicted by the fly. Protein-deprived female flies expressing CsChrimson in bitter taste neurons were subjected to such a probabilistic experimental design for an hour. Similarly to the experiments described in *Figure 4C and D*, for each fly red light was either triggered after 1.5, 3, or 6 s. As expected from *Figure 4D*, in stimulation trials, the length of activity bouts was shortened to different extents under the three different delays. Intriguingly however, in the catch trials, the food interaction bouts were much longer than in control experiments where no light was triggered throughout the experiment (*Figure 4F*). This effect was independent of the length of the interval between food contact and light onset.

This data is consistent with the hypothesis that flies undergoing the optogenetic activation protocol learn to expect that their interaction with the food will be 'interrupted' by a bitter stimulus after a relatively short time window. When this expectation is not met, the animal compensates by staying longer on the food, allowing it to ingest as much food as possible during the bout. An alternative explanation of why flies exhibit longer activity bouts might be due to a rebound effect: since activation of *Gr66a* neurons prevents consumption of sufficient food, flies remain hungry which leads them

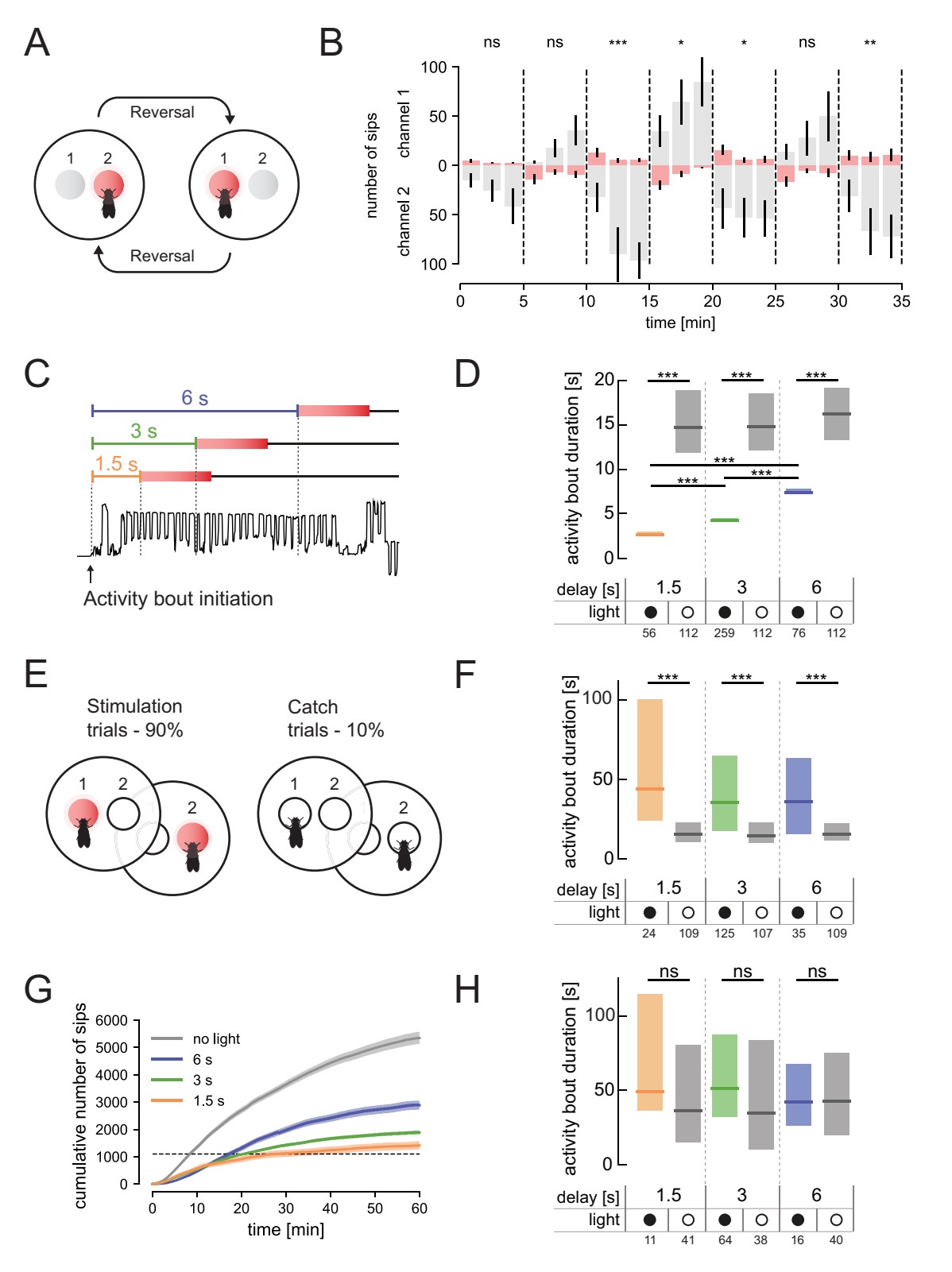

**Figure 4.** The optoPAD allows for complex dynamic closed-loop experimental designs. In all experiments, 5–7 days old female *Gr66a-GAL4 > CsChrimson* flies were used. (**A**) Schematic overview of the dynamic virtual taste reality experiment: every five minutes the contingency of the experiment is reversed (in each experimental block the fly's interaction with a different channel triggered light stimulation). (**B**) Number of sips from channel 1 (upper half of the plot) and channel 2 (lower half of the plot) across time in the changing virtual taste reality setting described in A. Columns
*Figure 4 continued on next page*

*Figure 4 continued*

and lines represent mean and the standard error of the mean, respectively. The trials leading to LED activation are shaded in red. (C) Onset of light stimulation (red box) can be freely set to occur at different times after the initiation of an interaction with food (delay of 1.5, 3 and 6 s). The lower part of the diagram represents a representative capacitance trace with the onset of food contact marked with an arrow. (D) Duration of activity bouts in flies exposed to light after different delays relative to the initiation of food interactions and corresponding controls (experimental design described in C). Plotted are the duration of activity bouts for the stimulated flies (light) and for the same number of trials that were longer than 1.5, 3 and 6 s (from left to right) performed by the 'no light' control flies. (E) Schematic of the experimental design in which light activation was set to happen in a probabilistic manner. (F) Duration of activity bouts of the catch trials. Plotted are the duration of activity bouts for the stimulated flies (light) and for a selection of 10% of all the trials that were longer than 1.5, 3 and 6 s (from left to right) performed by the 'no light' control flies. (G) Cumulative feeding for the four different groups of the experiment described in (E). Line represents the mean and the shading the standard error of the mean. Dotted line indicates the 1100 sips threshold used to calibrate the data for the internal state of the animal. (H) Duration of activity bouts of the catch-trials for sip-calibrated flies (trials performed until the flies had reached a total of 1100 sips). Plotted are the duration of activity bouts for the stimulated flies (light) and for a selection of 10% of all the trials that were longer than 1.5, 3 and 6 s (from left to right) performed by the 'no light' control flies. For genotypes, see Materials and methods and key resources table. ***$p<0.001$, **$p<0.01$, *$p<0.05$, ns non-significance. The numbers below the graphs in D, F and H indicate the number of flies tested in each condition. In D, F, and H, boxes represent median with upper/lower quartiles. In D, F and H, groups were compared by Kruskal-Wallis test, followed by Dunn's multiple comparison test. In B, the total number of sips for all bins in each channel during each period of 5 min was compared by Wilcoxon rank-sum test.

DOI: https://doi.org/10.7554/eLife.43924.021

The following source data and figure supplements are available for figure 4:

**Source data 1.** Source data file for *Figure 4B*.
DOI: https://doi.org/10.7554/eLife.43924.024
**Source data 2.** Source data file for *Figure 4D*.
DOI: https://doi.org/10.7554/eLife.43924.025
**Source data 3.** Source data file for *Figure 4F*.
DOI: https://doi.org/10.7554/eLife.43924.026
**Source data 4.** Source data file for *Figure 4G*.
DOI: https://doi.org/10.7554/eLife.43924.027
**Source data 5.** Source data file for *Figure 4H*.
DOI: https://doi.org/10.7554/eLife.43924.028
**Figure supplement 1.** Activity bouts of individual flies in a dynamic experimental protocol experiment.
DOI: https://doi.org/10.7554/eLife.43924.022
**Figure supplement 1—source data 1.** Source data file for *Figure 4—figure supplement 1*.
DOI: https://doi.org/10.7554/eLife.43924.023

to overconsume on catch trials. This hypothesis is supported by the analysis of the dynamics of food intake, which reveals clear differences in total food intake across the experimental groups (*Figure 4G*). These differences should lead to drastic differences in internal states between the flies exposed to the light and those exposed to the no-light control situation, and hence big differences in the feeding microstructure (*Corrales-Carvajal et al., 2016*; *Itskov et al., 2014*). In order to control for the effect of internal state on the catch trials, we decided to only include in our analysis the catch trials, in the time period until the flies have performed their first 1100 sips of yeast, therefore controlling for differences in food intake (dashed line in *Figure 4G*). This analysis revealed no significant differences in the length of food interactions in catch trials between the flies that did not receive *Gr66a* neurons stimulation (no light) and any of the experimental groups (*Figure 4H*). This result strongly suggests that the originally observed increase in the duration of food interactions during catch trials can be explained by differences in the deprivation state of the fly. This interpretation is supported by the observation that the main difference between the calibrated and non-calibrated data is the increase in the length of activity bouts in the non-light population of the calibrated dataset (*Figure 4H*). While we cannot rule out that in this behavioral paradigm flies learn to adapt their behavior to an expectation of a possible outcome, the behavioral effects can be explained by differences in internal metabolic state induced by alteration in the feeding behavior of the animal. Overall these experiments show that the optoPAD system allows the implementation of behavioral paradigms, in which stimuli are delivered optogenetically in a probabilistic manner. It however also highlights the importance of controlling for differences in internal state induced by complex experimental designs when interpreting behavioral data. Importantly, this challenge can be overcome using the rich and highly quantitative behavioral data generated by the flyPAD system.

# Discussion

We describe an experimental setup which allows the closed-loop optogenetic manipulation of specific neurons in *Drosophila melanogaster* contingent on the feeding behavior of the animal. The system extends the ability of the flyPAD to efficiently detect interactions of flies with food using capacitive sensing, by implementing real-time analysis of feeding behavior and the ability to activate light sources of different wavelengths depending on this behavior. This has allowed us to explore the effect of activating and silencing specific gustatory neurons on feeding behavior. Importantly, by activating and inhibiting sweet and bitter neurons, we were able to produce both phagostimulatory and phagoinhibiting effects in a coherent manner. This demonstrates that our closed-loop optogenetic manipulations are able to induce all phenotypes expected from the known biology of these neurons. Given that the exposure of the animal to gustatory stimuli is highly dependent on its behavior, it is essential that the design of gustatory circuit manipulations takes into account the behavior of the animal. We therefore envisage that the optoPAD will be especially valuable when exploring the involvement of sensorimotor circuits in feeding behavior. As the modular regulation of feeding microstructure plays a key role in nutrient homeostasis, the high temporal sensitivity of the flyPAD makes the optoPAD ideally suited to study its circuit basis.

The ability to analyze feeding behavior in a quantitative, high-resolution, and high-throughput fashion using the flyPAD technology has been leveraged to significantly advance our understanding of *Drosophila* feeding behavior. The optoPAD highlights a further advantage of this technology: its flexibility and expandability. After we described the first use of the flyPAD technology to manipulate behavior using optogenetics (*Steck et al., 2018*), Jaeger and colleagues also described an adaptation of the flyPAD technology to manipulate taste neurons during feeding (*Jaeger et al., 2018*). Interestingly, their approach is fundamentally different from the one we described, attesting to the flexibility of the flyPAD system. They define at the level of the hardware, specific features of the capacitance signal to trigger light activation. This approach allows the rapid triggering of light when the capacitance signal reaches a certain level, which is likely to be highly correlated with the time periods the fly touches the food. Given that the analysis of the behavior is implemented at the level of the hardware, it is however not possible for the user to readily modify the analysis of the behavior leading to the triggering of the light or to modify how and when the light stimulus gets triggered (delays, probabilities of stimulation, dynamic changes to the stimulation protocol). Furthermore, while this system is fast, the signal is not processed to detect sips and the light onset is therefore unlikely to reliably coincide with these feeding events. While our experiments clearly show that the current optoPAD system is able to modulate feeding behavior using closed-loop optogenetic manipulations, the ability to process the behavioral data in a flexible and complex way is likely to introduce a longer latency than that of the STROBE system. This latency can be counteracted using the existing knowledge of the very stereotyped parameters underlying feeding behavior, which allow us to tune the stimulation parameters in order for the light to better coincide with sip onset. It should also be possible to further optimize the efficiency with which the hardware handles the incoming signals and the speed at which the software processes these, leading to a significant improvement in the temporal precision of the delivery of the light stimulus. This might be important when dissecting the relationship between circuit activity and behavior at the scale of a single action potential. There will, however, always be a conflict between flexibility in experimental design, requiring computation time to analyze the behavioral signal, and minimizing the delay between behavior detection and light stimulation. We propose that as long as the delay is not beyond the behaviorally meaningful scale, the advantages of flexibility outweigh small improvements in the speed of the system. In any case as shown by both the optoPAD and the STROBE system, the flyPAD technology is flexible enough to allow the implementation of either priority.

Given its flexible design, the optoPAD system not only allows for a fixed closed-loop manipulation of neurons, but confers substantial flexibility to define how feeding behavior is linked to light activation. The researcher can for example predefine how many light stimulations will happen in one session, stimulate after the termination of an activity bout, or perform experiments in an open-loop mode. An important further feature if the possibility to design experiments in which light activation is altered to be probabilistic, while occurring at a fixed rate. This allows the experimenter to probe the effect of neuronal manipulations altering feeding behavior using interleaved control trials (catch trials). We expect that the ability to alter the dynamics and probabilities of optogenetic

manipulations will be most useful to manipulate the behavior of *Drosophila* in order to probe its ability to learn complex contingencies. By activating 'reward' and 'punishment' neurons in closed-loop using different behavioral contingencies, the optoPAD system should allow the design of new operant conditioning paradigms in which the animal learns to associate the consequences of its own behavior with specific outcomes or the statistical structure of its environment. Learning such abstract environmental structures is a fundamental ability animals use to optimize their foraging strategies (*Chittka, 2017*; *Glimcher and Fehr, 2013*). The extent to which *Drosophila* is able to perform such proto-cognitive computations is currently an important frontier in fly systems neuroscience. We expect that the flexibility of the optoPAD as well as its high-throughput design will allow researchers to explore how flies adapt their behavior to complex environmental features and identify the underlying neuronal circuits and computations.

# Materials and methods

### Key resources table

| Reagent type (species) or resource | Designation | Source or reference | Identifiers | Additional information |
|---|---|---|---|---|
| Genetic reagent (*D. melanogaster*) | *Gr5a-GAL4* | other | | Obtained from Kristin Scott lab |
| Genetic reagent (*D. melanogaster*) | *pUAS-Chrimson-mVenus* | BDSC | BDSC ID: 55136 | |
| Genetic reagent (*D. melanogaster*) | 57 F03-GAL4 | BDSC | BDSC ID: 46386 | |
| Genetic reagent (*D. melanogaster*) | *pBDP-GAL4Uw* | BDSC | BDSC ID: 68384 | |
| Genetic reagent (*D. melanogaster*) | *W[1118]* | other | | Obtained from Barry Dickson lab |
| Genetic reagent (*D. melanogaster*) | *UAS-GtACR1* | DOI: 10.1038/nmeth.4148 | | Obtained from Adam Claridge-Chang lab |
| Genetic reagent (*D. melanogaster*) | *Gr64f-GAL4* | BDSC | BDSC ID: 57668 | |
| Genetic reagent (*D. melanogaster*) | *attP2* | BDSC | BDSC ID: 8622 | |
| Genetic reagent (*D. melanogaster*) | *Gr66a-GAL4* | other | | Obtained from Bassem Hassan lab |
| Genetic reagent (*D. melanogaster*) | 67E03-GAL4 | BDSC | BDSC ID: 39441 | |
| Genetic reagent (*D. melanogaster*) | *SS02299-GAL4* | Janelia Research Campus; DOI: 10.7554/elife.34272 | Robot ID: 3018165 | |
| Software | Custom-written scripts in MATLAB R2013b | MathWorks | RRID: SCR_001622 | |
| Software | Bonsai 2.4 | https://bonsai-rx.org/, DOI: 10.3389/fninf.2015.00007 | | |

## Fly stocks and rearing conditions

Flies were reared at 25°C, 70% relative humidity (RH) in the dark to prevent non-specific activation of neurons. Flies were reared at standard density and were matched for age and husbandry conditions. The yeast-based fly medium (YBM) contained 80 g cane molasses, 22 g sugar beet syrup, 8 g agar, 80 g corn flour, 10 g soya flour, 18 g yeast extract, 8 ml propionic acid, and 12 ml nipagin (15% in ethanol), per liter. All data are from 24 hr wet-starved male flies unless otherwise stated. After hatching flies were aged for 5–7 days, then groups of 12–15 males were transferred to new vials and approximately 10 wild-type female flies were added. They were kept on YBM medium containing 0.4 mM all-trans-retinal (Sigma-Aldrich, #R2500, made using a stock solution of 100 mM all-trans-retinal dissolved in ethanol) for 2 days. To induce starvation, the flies were then transferred to vials containing tissue paper soaked with 5 ml Milli-Q water containing 0.4 mM all-trans-retinal 24 hr before the experiment. The starvation was chosen to increase carbohydrate appetite and to ensure robust feeding on sucrose.

For the experiments described in *Figure 1—figure supplement 1*, *Figure 2B*, *Figure 2—figure supplement 1*, *Figure 4* and *Figure 4—figure supplement 1* mated female flies were deprived of yeast for 3 days on a tissue soaked with 6.5 ml of 100 mM sucrose, and 0.4 mM all-trans-retinal.

## Hardware design and real-time data analysis

The optoPAD system is a new generation of the flyPAD, that was previously described in *Itskov et al. (2014)*. Additional hardware to the flyPAD was designed to fit on top of each of the 32 behavioral arenas and allow independent activation of 32 high-power (10 W) RGBA LEDs (ref. no. LZ4-00MA10; LED Engin, San Jose, CA, USA). Printed circuit boards were designed using Eagle CAD software (Cadsoft - version 6.2.0). The designs were then sent to Eurocircuits (http://www.euro-circuits.com/) for production.

The PCBs hosting the LED (LED board) measure 5 × 5 cm and fit exactly on top of the flyPAD arenas supported by screws and nuts that align the two boards. The LED is placed at the center of each board directly facing the middle of the flyPAD arena. An additional electrical circuit on each LED board includes four MOSFET N-ch (one for each LED color) that serve as switches for the LEDs and a set of five resistors connected in series to set the correct voltage to the pole of each individual color LED (3.3 and 0.5 Ω for red; 1 and 0.5 Ω for amber; 2 Ω for green, no voltage drop resistor was used for the blue color).

Three groups of twelve LED boards (3 × 12 = 36, four were not used) receive power and LED control signals from a control breakout board, which hosts the microcontroller (Arduino Mega 2560). These boards receive and distribute power from an external power supply unit (Corsair CSM 550W 80 + Gold Certified Semi-Modular ATX) through SATA power cables, allocating a maximum of 3.3 V to both amber and blue LEDs and of 5 V to both red and green LEDs. The microcontroller sends activation signals of 5 V to the correct transistor to be activated based on the information from serial communication with the computer. To alter the LED brightness, we used the Voltage dial of a switch mode bench power supply (Circuit Specialists CSI3060) to set the voltage applied to the LED.

All PCB designs can be found on the following GitHub repository: https://github.com/ribeiro-lab/optoPAD-hardware (*Goldschmidt, 2019b*; copy archived at https://github.com/elifesciences-publications/optoPAD-hardware).

## Real-time detection of food interactions and LED activation

The detection of food interactions in real time as well as the closed-loop control of LED illumination was performed using a custom-written Bonsai workflow (*Lopes et al., 2015*). This visual programming framework allows for real-time analysis of the flyPAD capacitance data sampled at 100 Hz and is capable of online communication with actuators for closed-loop experiments.

Unlike the data processing described in *Itskov et al. (2014)*, the optoPAD system detects when the fly interacts with a food source in real time. To do so, the Bonsai workflow continuously takes the absolute difference of two consecutive samples and applies a finite impulse response (FIR) filter with a running window of 50 samples. The filtered signal is finally thresholded with a value of 120 (a. u.), resulting in a binary signal representing if activity bouts were detected (*Figure 1C*). We chose the values for the running window size and threshold based on optimizing the accuracy of the

activity bout detection using the output of the offline detection algorithm as a standard (*Itskov et al., 2014*).

The protocol for LED activation implemented in Bonsai is a series of conditional nodes that can be programmed to perform flexible experiments.

Through this Bonsai workflow, each experiment can be programmed to run with the same protocol for as long as the user desires. The protocol used to control LED activation can be programmed independently for each flyPAD capacitance channel. Every time the fly starts interacting with food, a trial is initiated. From there, two possible behavioral outcomes are possible: 1. the fly stops interacting with food before the set 'delay' period (such activity bouts are classified as short trials, with no LED activation); 2. the fly keeps eating for longer than the 'delay' period. When the latter occurs, and depending on the experimental design, LED activation may or may not occur. This feature is dependent on the setting of the 'probability of stimulation' value. Bonsai pseudo-randomly samples a real number between 0 and 1 from a uniform distribution and compares it to a given previously defined probability. If this number is less or equal than the defined probability, the fly receives light stimulation for a period of time that is set by the user prior to the experiment. Else no stimulation occurs and the trial is considered a 'catch-trial', which only terminates when the fly stops interacting with the food at the end of the activity bout. Upon termination of light stimulation, the protocol is restarted, and the system will wait for the fly to start interacting with food again (if the fly continues to interact with food during the light stimulation, the ongoing activity bout is considered a new activity bout). Additionally, the user can define how many stimulations will occur on each channel.

For dynamic virtual taste environment experiments (*Figure 4A and B*), in which the channel controlling LED activation is changed every five minutes, a timer was added to the Bonsai workflow, which after a set time instructs the software to change the experimental parameters by reading from another configuration file.

Software can be found on the following GitHub repository: https://github.com/ribeiro-lab/opto-PAD-software (*Goldschmidt, 2019a*; copy archived at https://github.com/elifesciences-publications/optoPAD-software).

## Irradiance measurements

We performed the irradiance measurements shown in *Figure 2A* using an optical power meter (Thorlabs PM100D) and a standard photodiode power sensor (Thorlabs S121C). In order to accurately measure how much irradiance reaches the fly upon LED illumination, we placed the sensor at the same distance from the LED as the arena. We varied the voltage applied to the LED between 1.5 and 5 V in 0.5 V steps, and measured the peak value of power on the optical power meter upon LED illumination. Measurements were started at 1.5 V as no measurable optical power was detected below this voltage. We carried out each measurement three times. The irradiance was computed by dividing the optical power measurements by the effective sensor area ($14.923 \text{ mm}^2$).

## Behavioral experiments

Behavioral experiments were performed at 25°C, 70% RH. We used flies expressing either GtACR1 (*Mohammad et al., 2017*) or CsChrimson (*Klapoetke et al., 2014*) in subsets of gustatory neurons. The genotypes of the lines used in the manuscript are listed in the key resources table. optoPAD assays were performed following a protocol previously described (*Itskov et al., 2014*). Briefly, both wells of the optoPAD were filled with solutions containing different concentrations of sucrose (*Figures 2B* and *3A*: 5 mM; *Figure 3B*: 20 mM; *Figure 3C*: 100 mM; *Figure 3D*: 50 mM +10 mM quinine) or 10% yeast solution (*Figures 2C* and *4*). All solutions were in 1% agarose. Single flies were transferred to optoPAD arenas by mouth aspiration and allowed to feed for 1 hr in a light shielded box.

To test how different light intensities affect feeding behavior (*Figure 2B and C*), red (625 nm) LED activation was set to occur 0 s after the initiation of each activity bout for the experiment shown in *Figure 2B* and green (523 nm) LED activation was set to occur 0.5 s after each bout initiation for *Figure 2C*. For both experiments, light stimulation was sustained for 1.5 s independent of the flies' behavior. Based on the dose-response curves in *Figure 2*, we chose to use 3.5 V for the green LED and 2.25 V for the red LED for the experiments in *Figure 3*.

For the experiments aiming at inducing and repressing aversive and appetitive behaviors (*Figure 3*), LED activation was set to occur 0.01 s (*Figure 3A*), 0 s (*Figure 3B*), and 0.5 s (*Figure 3C and D*) after an activity bout was detected in the light-triggering channel. In *Figure 3A and C* the red LED was used and in *Figure 3B and D* the green LED was used. Light stimulation was sustained for 1.5 s independent of the behavior of the fly.

For the dynamic virtual taste environment experiment (*Figure 4A and B*), red LED activation was set to occur 0.5 s after the initiation of each activity bout in the light-triggering channel (channel 1 or two depending on the experimental block), and the light activation was sustained for 2 s independently of the activity of the fly. For the 'control' channel, interactions of the fly with the food did not trigger any LED activation.

For the experiments testing the flies' behavioral response to activation of bitter-sensing neurons following a delay in relation to the initiation of the activity bout (*Figure 4C–H*), the red LED was set to occur 1.5, 3 or 6 s after an activity bout was detected in 90% of the trials on both food sources. Light stimulation was sustained for 2 s. A new trial was initiated when a new interaction with food was detected. The 10% of trials with no LED activation were terminated at the end of the respective activity bout.

## Statistics

Results of optoPAD experiments were compared using the Kruskal-Wallis test, followed by Dunn's multiple comparison test when more than two groups were compared or Wilcoxon rank-sum test, followed by Bonferroni correction when multiple comparisons were made. All tests were two-tailed.

## Acknowledgements

We thank Gwyneth Card, Richard Benton, and Adam Claridge-Chang for providing fly strains. Lines obtained from the Bloomington Drosophila Stock Center (NIH P40OD018537) were used in this study. We thank Gonçalo Lopes, João Frazão, Niccolò Bonacchi, Filipe Carvalho and Artur Silva for help in software and hardware design. We thank Daniel Münch, and all members of the Behavior and Metabolism laboratory for helpful discussions and comments on the manuscript. This project was supported by the Portuguese Foundation for Science and Technology (FCT) postdoctoral fellowship SFRH/BPD/79325/2011 to PMI and doctoral fellowship PD/BD/114273/2016 to DG; the Human Frontier Science Program Project Grant RGP0022/2012; the BIAL Foundation Grants (283/14 and 279/16). Research at the Centre for the Unknown is supported by the Champalimaud Foundation.

## Additional information

### Competing interests

Pavel M Itskov: has a commercial interest in the flyPAD open-source technology. The other authors declare that no competing interests exist.

### Funding

| Funder | Grant reference number | Author |
|---|---|---|
| Fundação para a Ciência e a Tecnologia | SFRH/BPD/79325/2011 | Pavel M Itskov |
| Fundação Bial | 283/14 | Carlos Ribeiro |
| Fundação Bial | 279/16 | Carlos Ribeiro |
| Fundação para a Ciência e a Tecnologia | PD/BD/114273/2016 | Dennis Goldschmidt |
| Champalimaud Foundation | | Carlos Ribeiro |

The funders had no role in study design, data collection and interpretation, or the decision to submit the work for publication.

## Author contributions

José-Maria Moreira, Resources, Data curation, Software, Formal analysis, Validation, Investigation, Visualization, Methodology, Writing—review and editing; Pavel M Itskov, Conceptualization, Resources, Data curation, Software, Formal analysis, Supervision, Visualization, Methodology, Writing—original draft, Writing—review and editing; Dennis Goldschmidt, Conceptualization, Data curation, Software, Formal analysis, Validation, Investigation, Visualization, Methodology, Writing—original draft, Project administration, Writing—review and editing, Supervision; Celia Baltazar, Validation, Investigation, Writing—review and editing; Kathrin Steck, Data curation, Formal analysis, Validation, Investigation, Visualization, Methodology, Writing—review and editing; Ibrahim Tastekin, Formal analysis, Validation, Investigation, Writing—review and editing, Supervision; Samuel J Walker, Formal analysis, Validation, Investigation, Methodology, Writing—review and editing; Carlos Ribeiro, Conceptualization, Formal analysis, Supervision, Funding acquisition, Visualization, Methodology, Writing—original draft, Project administration, Writing—review and editing

## Author ORCIDs

José-Maria Moreira (iD) https://orcid.org/0000-0003-2420-7930
Pavel M Itskov (iD) https://orcid.org/0000-0002-2191-7083
Dennis Goldschmidt (iD) https://orcid.org/0000-0002-0603-7176
Kathrin Steck (iD) http://orcid.org/0000-0003-2711-2873
Samuel J Walker (iD) https://orcid.org/0000-0003-3118-8467
Carlos Ribeiro (iD) https://orcid.org/0000-0002-9542-7335

## Decision letter and Author response

Decision letter https://doi.org/10.7554/eLife.43924.031
Author response https://doi.org/10.7554/eLife.43924.032

# Additional files

## Supplementary files

• Transparent reporting form
DOI: https://doi.org/10.7554/eLife.43924.029

## Data availability

Data used to generate the plots in the figures are included in the manuscript and supporting files. The optoPAD hardware designs and software are made available on GitHub at https://github.com/ribeiro-lab/optoPAD-hardware and https://github.com/ribeiro-lab/optoPAD-software, respectively. Copies have been archived at https://github.com/elifesciences-publications/optoPAD-hardware and https://github.com/elifesciences-publications/optoPAD-software.

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
