## [Decision Letter]

Thank you for submitting your article "optoPAD: a closed-loop optogenetics system to study the circuit basis of feeding behaviors" for consideration by *eLife*. Your article has been reviewed by K VijayRaghavan as the Senior Editor, Mani Ramaswami as the Reviewing Editor, and three reviewers. The reviewers have opted to remain anonymous.

The reviewers have discussed the reviews with one another and the Reviewing Editor has drafted this decision to help you prepare a revised submission.

Summary:

The authors recently published a paper describing the use of an automated food choice assay flyPAD to study yeast taste neurons and their effects on yeast feeding. The current manuscript serves as a technical follow-up in which Moreira et al. develop and validate a closed-loop optogenetics system, optoPAD, built on top of the previously described flyPAD. OptoPAD produces light stimulation time-locked to the feeding onset detected by flyPAD, allowing for temporally precise and dynamic manipulation of targeted neuron populations. Using Bonsai data stream processing, they first extract active periods when fly is interacting with the food source. This information is then used to control high power LEDs in different wavelengths at precise time points and durations to optogenetically activate specific classes of neurons. The authors demonstrate the efficacy of their system by stimulating sugar feeding by activating sugar taste neurons or reducing yeast feeding by inhibiting taste peg neurons. Next, they also perform a dynamic task to change the acceptance of a food source in a two-choice assay by pairing one source with bitter taste neuron activation. These experiments show that close loop optoPAD system can change the value of a food source in real time by means of optogenetic activation of the avoidance pathways in the periphery. Thus, this system is shown to be useful and effective in creating virtual taste realities while individual flies are freely feeding on a define food medium. Overall, the work establishes that optoPAD system has a potential to contribute to the understanding of sensory responses and reinforcement learning in flies.

In general, the manuscript is well organized and clearly explained. However, the authors do not fully exploit the system to demonstrate its potential utility for a broad range of *Drosophila* applications that would greatly expand the interest and impact of this paper.

Essential revisions:

To be acceptable for publication, the manuscript must be revised with new data to: (a) show that optoPAD's utility is not limited to peripheral neurons and that can be useful for identifying/manipulating central neurons, and (b) include options that allow optoPAD to be flexible enough to allow optogenetic stimulations at times other than the onset of feeding.

1) In Figure 1, authors describe the real-time detection of food interactions in flyPAD. What is the false positive and false negative rate of this real-time data analysis? How reliable is the thresholding to detect food interactions? It would be good to show confidence rates of the system by annotating files by a human observer and comparing it with the machine annotations.

2) Authors measure the latency of the optoPAD to trigger the LED illumination and found this delay is 50-120ms. Could they mention why the delay has a 70ms range? What causes the variability? They also mention average sip duration is 130ms. Would this mean if the bout length is lower than 50-120ms, there will be a chance LED would not be activated before the sip ends? How would this impact fly's behavior? Although 10ms delay probably would not directly impact behavior, it may influence the action potentials in neurons that regulate the behavior in time periods smaller than 10ms. Could authors discuss these possibilities in their Discussion rather than claiming the delay will not impact behavior without actually testing it?

3) All of the transgenic fly strains tested in the optoPAD experiments mainly label taste sensory neurons that stimulate or inhibit feeding. Does optoPAD work for central feeding circuits to modulate feeding behavior? Testing activation of NPF or sNPF neurons that are shown to regulate food responses can demonstrate functionality of this system in neurons of the central nervous system rather than periphery. Would optoPAD work with a restricted driver that label few neurons? What are the limits of this system?

4) On the same lines, a potential advantage of the optoPAD system is to allow the design of operant conditioning experiments to study "pleasure circuits", which is challenging in flies. To do this, it would be ideal to be able to trigger optogenetic stimulation in response to touching the food or licking the food. It has previously been shown that activation of NPF neurons is in itself rewarding (Shohat-Ophir et al., 2012). Thus, it would be interesting to test whether manipulating NPF neuron activity can support operant conditioning paradigms using the optoPAD system.

5) In addition to showing that the system can modulate central neurons, can the authors determine if the system is suited for the dynamic regulation of more complex neural modules that control the microstructures of fly feeding responses to various food?

6) In Figure 4, authors try an elegant experiment to change the value of a food source by activating bitter taste neurons in a two-choice assay. They see no difference between two sides in the first 10 minutes of the experiment even though one side is paired with punishment (bitter neuron activation). Can authors explain the 10-minute delay in the results further? In this period of time are the flies interacting with the food at all?

7) To expand the versatility of optoPAD, it would be useful to add an "open-loop" mode in the setup, where light stimulation can be controlled at will and does not need to follow feeding onset. This will allow users to take full advantage of the high temporal resolution of both flyPAD and optogenetics to study feeding related modalities before ingestion happens, such as tasting and feeding initiation.

8) To allow for more diverse experimental designs, it would be useful if one could choose to deliver the stimulations at the end of a feeding bout. For example, a researcher might be particularly interested in the inter-meal interval. In this case, stimulation placed at the end of an activity bout will help to evaluate whether circuit manipulation affects the latency to start next feeding bout since the cessation of the last one.

9) It would also be useful to be able to control the number of optogenetic stimuli delivered to the fly, for example, if one wanted to have a maximum number of activations for the targeted neural circuit.

---

## [Author Response]

Essential revisions:To be acceptable for publication, the manuscript must be revised with new data to: (a) show that optoPAD's utility is not limited to peripheral neurons and that can be useful for identifying/ manipulating central neurons, and (b) include options that allow optoPAD to be flexible enough to allow optogenetic stimulations at times other than the onset of feeding.

We have significantly expanded the software used to control the contingency of the optogenetic activation and added new experiments showing that we can manipulate central neurons. We describe in detail how we addressed these requested revisions in our point by point answer to the reviewers. But we would like to give an overview of how we addressed these already here.

Regarding point (a): We now include data in which we use the optoPAD system to manipulate escape neurons leading to alterations in feeding. These are a small set (2) of centrally located neurons deep within the brain of the fly. We chose them as we wanted a set of neurons which are (a) very sparse, (b) very deep within the brain, (c) have a clear behavioral phenotype which can be seen independently from the flyPAD readout, and (d) should lead to a reliable termination of feeding when activated. Indeed, activating these neurons using CsChrimson after the fly initiates feeding led to a strong reduction in feeding behavior on the food source paired with light. These new data clearly show that our system is not limited to inducing changes in feeding behavior upon manipulation of peripheral neurons but can also be used to induce changes in feeding behavior upon optogenetic manipulation of sparse and centrally located neurons.

Regarding point (b): We have now modified the scripts controlling the light stimulus to significantly expand the conditions upon which the light stimulus can be delivered. More specifically it is now possible to: (i) trigger light activation in an open loop fashion, meaning in a predefined pattern, not contingent on the behavior of the animal; (ii) trigger light activation when the fly stops interacting with the food, and; (iii) we have included a counter which allows the user to predefine the number of times the light stimulus is triggered, therefore limiting the number of stimulations the animal experiences.

1) In Figure 1, authors describe the real-time detection of food interactions in flyPAD. What is the false positive and false negative rate of this real-time data analysis? How reliable is the thresholding to detect food interactions? It would be good to show confidence rates of the system by annotating files by a human observer and comparing it with the machine annotations.

In our paper describing the development of the flyPAD system (Itskov et al., 2014) we performed a thorough comparison of the behavioral classification of feeding behavior by our post-hoc offline algorithms with that of a manual annotator. This comparison allowed us to show that the offline algorithm had very low false positive and false negative rates. As suggested by the reviewer we now tested the reliability of the real-time data analysis performed by the optoPAD system. To do so we compared the performance of the new online algorithm with the previously validated offline algorithm. This analysis revealed that our online algorithm misses to classify 8.5% of capacitance samples as activity bouts (false negatives) and only misclassifies 1.6% of samples as activity bouts (false positives). These data are within the accuracy obtained for similar state-of-the-art methods. We describe the approach and the obtained data in subsection “The optoPAD system” of the revised manuscript.

2) Authors measure the latency of the optoPAD to trigger the LED illumination and found this delay is 50-120ms. Could they mention why the delay has a 70ms range? What causes the variability? They also mention average sip duration is 130ms. Would this mean if the bout length is lower than 50-120ms, there will be a chance LED would not be activated before the sip ends? How would this impact fly's behavior? Although 10ms delay probably would not directly impact behavior, it may influence the action potentials in neurons that regulate the behavior in time periods smaller than 10ms. Could authors discuss these possibilities in their Discussion rather than claiming the delay will not impact behavior without actually testing it?

We were indeed surprised to see that the delay in light activation had a range of 70 ms. We do not know the exact reason for this range. There is however a straight-forward possible explanation when one considers the hardware design of the flyPAD system. As explained in the text the observed range can be easily explained by the size of the buffer of the FPGA. The FPGA receives the data from the capacitance to digital converters and stores them temporarily before sending the data to the computer. When the data which will trigger the light activation reach the FPGA they will be either sent to the computer earlier, if the buffer already contains data, or later, if the buffer is relatively empty. Therefore, the buffer size of the FPGA could explain the observed range in the onset of the LED illumination. Given that the FPGA is not an essential part of the flyPAD design one can imagine designing a special flyPAD version which does not use an FPGA, therefore eliminating this possible source of delay.

We are sorry that we did not explain well why we think that even if the system has a delay in light onset, this should not lead to the fly feeding without receiving light stimulation. The main argument is that there is a delay between the fly touching the food (onset of an activity bout) and when the fly starts feeding (onset of a feeding burst). As you can see in the plot in Figure 1—figure supplement 1 this delay is quite reproducibly in the 400 ms bin. Therefore if the delay between detection of the activity bout and the light onset is set to 0 ms and if one assumes the worst case scenario in which the light onset is delayed by 120 ms after the detection of the activity bout, then light activation would still occur 280 ms before the animal starts feeding. It is therefore highly unlikely that the delay in the system leads to the fly feeding without experiencing the optogenetic stimulation. Furthermore, given that the timing between onset of feeding and contact with the food is so reproducible, by setting the delay between the detection of the behavior and light onset to 250 ms one can even time the light onset even more precisely to the onset of feeding. This is only possible because the optoPAD allows the user to define the delay between the detection of the activity bout and light onset.

3) All of the transgenic fly strains tested in the optoPAD experiments mainly label taste sensory neurons that stimulate or inhibit feeding. Does optoPAD work for central feeding circuits to modulate feeding behavior? Testing activation of NPF or sNPF neurons that are shown to regulate food responses can demonstrate functionality of this system in neurons of the central nervous system rather than periphery. Would optoPAD work with a restricted driver that label few neurons? What are the limits of this system?

We agree with the reviewers that formally showing that the optoPAD can be used to manipulate centrally located neurons to study their impact on feeding would make an important point. As suggested, we have performed experiments in which we used CsChrimson to stimulate neurons labeled by *NPF-Gal4* and *sNPF-Gal4*. Unfortunately, we could not detect an increase in feeding or other feeding related parameters using *NPF-Gal4*. When using *sNPF-Gal4* we could detect an increase in the time during which the flies interacted with the food, but we could also not observe a clear increase in feeding. Furthermore, the observed change in behavior was not limited to the food source triggering the light stimulation but was also visible on the control food source. This indicates that the simulation of *sNPF* neurons leads to the release of a factor (likely sNPF) which increases the interaction with food over longer time-scales, therefore making the stimulation effect nonspecific to the food source triggering the activation of the neurons. Finally, the effect of stimulation of sNPF neurons was not fully penetrant. We performed the experiment in 4 separate cohorts and failed to see it in one cohort.

To clearly show that we can induce changes in behavior by manipulating a very small set of centrally located neurons we therefore decided to manipulate the giant fiber neurons in the fly brain, which trigger escape behavior. These are a small set (2) of centrally located neurons deep within the brain of the fly. We chose them as we wanted a set of neurons which are (a) very sparse, (b) very deep within the brain, (c) have a clear behavioral phenotype which can be scored independently from the flyPAD readout, and (d) should lead to a reliable termination of feeding when activated. Indeed, activating these neurons using CsChrimson after the fly initiates feeding led to a strong reduction in feeding behavior on the food source paired with light. We now show these data in Figure 2—figure supplement 2 and discuss them in subsection “Optogenetic gustatory virtual realities”. These new data clearly show that our system is not limited to inducing changes in feeding behavior upon manipulation of peripheral neurons but can also be used to induce changes in feeding behavior upon optogenetic manipulation of sparse and centrally located neurons.

4) On the same lines, a potential advantage of the optoPAD system is to allow the design of operant conditioning experiments to study "pleasure circuits", which is challenging in flies. To do this, it would be ideal to be able to trigger optogenetic stimulation in response to touching the food or licking the food. It has previously been shown that activation of NPF neurons is in itself rewarding (Shohat-Ophir et al., 2012). Thus, it would be interesting to test whether manipulating NPF neuron activity can support operant conditioning paradigms using the optoPAD system.

We agree with the reviewers that the prospect of performing experiments to study the impact of “pleasure circuits” using operant conditioning experiments is an exciting future avenue of research using the optoPAD. As suggested, we have performed experiments in which we used CsChrimson to stimulate neurons labeled by *NPF-Gal4* and *sNPF-Gal4*. As mentioned above, unfortunately we could not detect an increase in feeding using *NPF-Gal4*. When using *sNPF-Gal4* we could detect an increase in the length during which the flies interacted with the food, but we could also not observe a clear increase in feeding. Furthermore, the observed change in behavior was not limited to the food source triggering light stimulation but was also visible on the control food source. This indicates that the simulation of sNPF neurons leads to changes over longer time scales therefore making the stimulation effect nonspecific to the precise action one would try to reinforce. While we therefore believe that it should be possible to reinforce specific behaviors (especially feeding) using the optoPAD, such experiments will need to be very carefully planned and include many different controls to ensure that it was possible to indeed operantly reinforce a specific behavior. Unfortunately, such experiments will require considerably more time than available to revise the manuscript and would also go beyond the scope of the current manuscript.

5) In addition to showing that the system can modulate central neurons, can the authors determine if the system is suited for the dynamic regulation of more complex neural modules that control the microstructures of fly feeding responses to various food?

We agree with the reviewers that the prospect of modulating neuronal populations controlling specific aspects of the feeding microstructure is a further exciting avenue for the use of the optoPAD system. Given our interest in that topic it was actually a key motivation for its development. Our use of the optoPAD to show that the taste peg neurons specifically control the length of the feeding bursts on yeast is in our opinion a powerful demonstration of this strategy (Steck et al., 2018). Unfortunately, at this stage there are to our knowledge no published central neurons which control specific aspects of the microstructure of feeding to various foods. This precludes us from performing the experiments suggested by the reviewers. We are however currently using the optoPAD system to successfully characterize neurons for feeding microstrucuture-specific phenotypes in ongoing projects in the laboratory.

6) In Figure 4, authors try an elegant experiment to change the value of a food source by activating bitter taste neurons in a two-choice assay. They see no difference between two sides in the first 10 minutes of the experiment even though one side is paired with punishment (bitter neuron activation). Can authors explain the 10-minute delay in the results further? In this period of time are the flies interacting with the food at all?

We agree with the reviewers that this was an intriguing observation. We therefore discussed that in the original version of the manuscript and speculated that this could be due to the flies having to acclimatize to the chamber and not showing a high level of interaction with the food during that time. This is something we had observed earlier using video tracking (Corrales-Carvajal et al., 2016).

To support this hypothesis we generated a raster plot of the behavior of every fly in the dynamic assay and quantified the average time every fly interacts with food over the whole period of the assay (Figure 4—figure supplement 1). While flies interact with the food during the first two 5-minute blocks, most flies show few and short bouts of interaction. However, importantly in all experimental blocks the flies always interact for longer periods of time with the non-stimulated (not virtual bitter) side of the food. It is also interesting to see that, except for the first block and the fourth block, in the first bin after the inversion the flies interact more with the previously inert food which now however leads to bitter neuron stimulation. Finally, these data indicate that in contrary to what we normally observe when deprived flies eat on yeast, flies undergoing the dynamic protocol hardly interact with the food after 25 minutes indicating that the dynamic feature of the environment “discourages” the flies from continuing to exploit this environment.

This analysis clearly indicates that the population of flies adapts their feeding preference to the current optogenetic contingency. The reason that the effect in the main figure is not significant for the first two experimental blocks is most likely due to the lack in statistical power to validate these observed effects.

7) To expand the versatility of optoPAD, it would be useful to add an "open-loop" mode in the setup, where light stimulation can be controlled at will and does not need to follow feeding onset. This will allow users to take full advantage of the high temporal resolution of both flyPAD and optogenetics to study feeding related modalities before ingestion happens, such as tasting and feeding initiation.

We thank the reviewer for this suggestion. We now include a script which the optoPAD to run in an “open-loop” mode. We also provide data in which we show that silencing taste-pegs neurons using the “open-loop” mode leads to the decrease in the length of the feeding burst on yeast (Figure 2—figure supplement 1). We describe these experiments in subsection “Optogenetic gustatory virtual realities”.

8) To allow for more diverse experimental designs, it would be useful if one could choose to deliver the stimulations at the end of a feeding bout. For example, a researcher might be particularly interested in the inter-meal interval. In this case, stimulation placed at the end of an activity bout will help to evaluate whether circuit manipulation affects the latency to start next feeding bout since the cessation of the last one.

We thank the reviewers for this suggestion. We have now integrated the ability to deliver the light stimulation after the termination of an activity bout in the newest release of the optoPAD software. We mention this extra capability in the Discussion section.

9) It would also be useful to be able to control the number of optogenetic stimuli delivered to the fly, for example, if one wanted to have a maximum number of activations for the targeted neural circuit.

We thank the reviewers for these suggestions. We have now integrated the requested counter in the newest release of the optoPAD software. We mention this extra capability in the Discussion section.